# CFD Study of the Effect of the Angle Pattern on Iliac Vein Compression Syndrome

**DOI:** 10.3390/bioengineering10060688

**Published:** 2023-06-05

**Authors:** Hsuan-Wei Chen, Chao-Hsiang Chen, Yu-Jui Fan, Chun-Yu Lin, Wen-Hsien Hsu, I-Chang Su, Chun-Li Lin, Yuan-Ching Chiang, Haw-Ming Huang

**Affiliations:** 1Graduate Institute of Biomedical Materials and Tissue Engineering, Taipei Medical University, Taipei 11031, Taiwan; chen31729@gmail.com; 2Department of Imaging Medicine, Taipei Medical University Hospital, Taipei 11031, Taiwan; sloveq0909@gmail.com; 3School of Biomedical Engineering, Taipei Medical University, Taipei 11031, Taiwan; ray.yj.fan@tmu.edu.tw; 4Department of Radiology, Tri-Service General Hospital, National Defense Medical Center, Taipei 114202, Taiwan; ginse1586@gmail.com; 5Department of Lymphovascular Surgery, Taipei Municipal Wanfang Hospital, Taipei 11600, Taiwan; angiohsu@gmail.com; 6Department of Surgery, School of Medicine, College of Medicine, Taipei Medical University, Taipei 11031, Taiwan; ichangsu@gmail.com; 7Department of Neurosurgery, Taipei Medical University-Shuang Ho Hospital, Ministry of Health and Welfare, New Taipei City 235041, Taiwan; 8Medical Device Innovation and Translation Center, Department of Biomedical Engineering, National Yang Ming Chiao Tung University, Taipei 112304, Taiwan; cllin2@nycu.edu.tw; 9Department of Mechanical Engineering, Chinese Culture University, Taipei 111396, Taiwan; jyj3@ulive.pccu.edu.tw; 10School of Dentistry, Taipei Medical University, Taipei 11031, Taiwan

**Keywords:** iliac vein, IVCS, computational fluid dynamics, finite element, angle pattern

## Abstract

Iliac vein compression syndrome (IVCS, or May–Thurner syndrome) occurs due to the compression of the left common iliac vein between the lumbar spine and right common iliac artery. Because most patients with compression are asymptomatic, the syndrome is difficult to diagnose based on the degree of anatomical compression. In this study, we investigated how the tilt angle of the left common iliac vein affects the flow patterns in the compressed blood vessel using three-dimensional computational fluid dynamic (CFD) simulations to determine the flow fields generated after compression sites. A patient-specific iliac venous CFD model was created to verify the boundary conditions and hemodynamic parameter set in this study. Thirty-one patient-specific CFD models with various iliac venous angles were developed using computed tomography (CT) angiograms. The angles between the right or left common iliac vein and inferior vena cava at the confluence level of the common iliac vein were defined as α1 and α2. Flow fields and vortex locations after compression were calculated and compared according to the tilt angle of the veins. Our results showed that α2 affected the incidence of flow field disturbance. At α2 angles greater than 60 degrees, the incidence rate of blood flow disturbance was 90%. In addition, when α2 and α1 + α2 angles were used as indicators, significant differences in tilt angle were found between veins with laminar, transitional, and turbulent flow (*p* < 0.05). Using this mathematical simulation, we concluded that the tilt angle of the left common iliac vein can be used as an auxiliary indicator to determine IVCS and its severity, and as a reference for clinical decision making.

## 1. Introduction

Iliac venous compression (IVC) is caused by the compression of the iliac vein by the upper arteries and the lower spine, which results in a reduction in the diameter of the blood vessels [1,2]. When IVC occurs, the venous hemodynamics change. This phenomenon may induce iliac venous compression syndrome (IVCS), wherein blood flow obstruction results in deep vein thrombosis or venous hypertension in the lower extremities [3,4]. IVCS is most commonly found at the junction of the right and left common iliac vein. In this position, the right common iliac artery crosses anterior to the left common iliac vein and runs down the right side of the pelvis. Since in humans the right iliac vein and artery do not have overlapping anatomical relationships, IVCS usually occurs only in the left iliac vein [4,5,6,7]. This compression may lead to venous return disorder and increased venous pressure in the lower extremities that may cause venous thrombosis and pain at the lower extremities [4,7,8,9].

Although the compression of the common iliac vein is common, there remains controversy over the condition’s prevalence, with clinical data from different investigators showing rates varying from 22 to 49% [4,8,10]. Additionally, only a small number of patients with IVC are symptomatic [4,9,10]. Although the true prevalence of IVCS remains unknown [4,9,10], several clinical reports have indicated that IVCS predominantly affects females [9,10,11,12], with the prevalence in women (24.1%) being much higher than that in men (2.7%) [8,9]. This clinical phenomenon may be due to the smaller average vessel diameters and greater lumbar curvature in females [3,13,14]. Although not life threatening, quality of life is reduced for symptomatic IVCS patients [4].

Although IVCS is caused by the compression of the iliac vein, the degree of stenosis that can indicate clinically significant observations is unclear [7]. IVCS diagnosis has, for the most part, been based on phlebography [4,9,15,16]. However, investigations of this method have found that IVCS is often improperly diagnosed [4,7,9,10]. Ultrasonography is another tool used to diagnose IVCS, but scholars have indicated that this method is both less sensitive [17] and subject to significant limitations [10,18]. A new technique, intravascular ultrasonography, has been recently reported to better detect IVCS, but expensive facilities and high patient costs have made this technology available only in a limited number of cases [10]. Finding diagnostic parameters to prevent misdiagnosis and inappropriate IVCS treatment remains an issue for physicians who treat this syndrome.

Though IVCS was first described over one hundred years ago, the condition still has no standardized diagnostic criteria [9]. Some clinical studies have reported that the critical level of blood flow turbulence occurs when the ratio of blood flow velocity before and after the vein compression site reaches 1:2.5 [4,5,19]. Other studies have indicated that a pressure gradient of more than 2 or 3 mmHg is an important marker to confirm IVCS [4,7,15,18]. One study defined a flow volume in the right common iliac vein 40% greater than left side as a critical cutoff point for IVCS [4]. In addition, studies have shown that the anatomical morphology of the iliac vein plays an important role in the occurrence of symptoms after compression [8,20]. Changes in vessel diameter at vein compression sites have been suggested as an important factor in some studies [8,21,22,23]. While the mean diameter of the left common iliac vein without compression varies from 7.5 mm to 13.1 mm, a reduction below 4 mm can be used as a diagnostic cutoff [8]. Liu et al. [23] suggested that stenosis in the luminal diameter greater than 50% could be an indicator of IVCS. Recent research has also indicated that the angle at the confluence between the left and right common iliac veins plays an important role in the syndrome [3,20]. For example, Li et al. [3] measured the angle between the two common iliac veins and inferior vena cava at the confluence level to set up a cutoff value for IVCS diagnosis to improve clinical decision-making. They found that this angle contributed to an increased prevalence of IVCS. However, the mechanism underlying how iliac vein morphology affects the occurrence of IVCS has not yet been determined.

Hemodynamics, the study of flow changes in the blood using physiology and fluid dynamics, provides quantitative information such as flow resistance, velocity, and pressure that can be used for diagnosing and treating cardiocirculatory system diseases [24,25]. Hemodynamics can be studied using computational fluid dynamics (CFD) [26], and recent developments in computers and medical imaging software have made the CFD analysis of blood flow both reliable and accessible to clinicians. CFD has been applied to analyze various vascular diseases, such as coronary artery disease [27], aortic dissection [19,28,29], liver radioembolization [30], and intracranial aneurysm [31].

Because of the difficulty of diagnosing IVCS, this study aimed to analyze the blood flow field in compressed iliac veins under various confluence angles between the left and right common iliac veins, with the hypothesis that the tilt angle of the left common iliac vein affects IVC severity in patients.

## 2. Materials and Methods

### 2.1. Patient Selection and Model Creation

The patient data acquisition and experimental methods of this study were approved by the TMU-Joint Institutional Review Board (N202107090) at Taipei Medical University. All patient images were obtained from clinically indicated computed tomography (CT) angiograms at Taipei Medical University. Patient variables for selected subjects are listed in Table 1. Figure 1a shows a CT image of a patient with a normal iliac vein angle as defined by Li et al. [3] and confirmed IVCS. CT images were converted to a series of DICOM (Digital Imaging and Communication in Medicine)-formatted 2D images. Each 2D image represented a layer of the venous vessel. The commercial 3D reconstruction software Materialise Mimics (Materialise NV, Leuven, Belgium) was employed to convert the 3D anatomical models to digital files. To fix the common geometric errors that occur when digital models are converted to CAD files, files were converted using Geomagic^®^ (3D Systems, Valencia, CA, USA) to surface patch files that could be imported to 3D CAD design software. SolidWorks (Dassault Systèmes S.A., Vélizy-Villacoublay, France) was then used to convert the geometrical shape to a solid model for further CFD analysis. In Figure 1b, the three-dimensional patient-specific geometries of the model contain both the left and right common iliac veins that come together to form the inferior vena cava. A significant compression can be observed on the left common iliac vein. Auto-meshing was performed prior to CFD analysis (Figure 1c). The model comprised approximately 1.7 million elements, with an element size of approximately 12 μm. With this mesh density, mesh convergence was achieved [32].

### 2.2. Boundary Conditions and Material Properties

In this study, CFD analysis was performed using the commercial finite element software package ANSYS Workbench 18.2 (Swanson Analysis Systems, Inc., Canonsburg, PA, USA). To calculate flow fields and hemodynamic parameters, blood flow in the iliac veins assumed incompressible Newtonian flow [25]. Values for blood fluid density and dynamic viscosity (1050 kg/m^3^ and 0.003 kg/m/s, respectively) were used according to previous studies [32,33,34,35]. As in previous CFD analyses, the model assumed venous wall rigidity [34]. The inlet blood velocities of the left (VL) and right (VR) iliac veins and inferior vena cava (VI) were 0.03 m/s, 0.12 m/s, and 0.06 m/s, respectively, determined by intravenous digital subtraction angiography (IVDSA) as reported previously [36,37]. Briefly, when injecting contrast into vessels, blood and contrast medium were mixed proportionally to their respective flow rates. Angiographic datasets were sent to a workstation running commercial software (Syngo iflow, Siemens Healthcare, Forchheim, Germany). Blood flow velocity was then calculated from contrast pixel intensity changes over time (Figure 2). The model was calculated and post-processed using the ANSYS package. 

### 2.3. Mathematical Modeling

Blood flow passing through a stenosis may produce turbulence with varying characteristics depending on the degree of stenosis. It is therefore important to use an appropriate turbulence model to calculate the flow field and obtain the hemodynamic parameters. Two turbulence models frequently used to study disturbance at the post stenotic region in arteries are the k-ω and k-ε turbulence models [38]. In this study, we first investigated the ability of these two models to predict flow behavior in venous stenoses.

The Reynolds-averaged Navier–Stokes (RANS) equations were used as flow governing equations. Firstly, the continuity and Navier–Stokes (NS) equations are:(1)$$\frac{\partial u_{i}}{\partial x_{i}}=0$$
(2)$$\rho \left(\frac{\partial u_{i}}{\partial t}+u_{j}\frac{\partial u_{i}}{\partial x_{j}}\right)=-\frac{\partial p}{\partial x_{i}}+\mu \frac{\partial^{2}u_{i}}{\partial x_{i}\partial x_{j}}$$
where *ρ* is fluid density, *u_i_* is velocity, *p* is pressure, *μ* is viscosity, and $x_{i}=\left(x,y,z\right)$ are Cartesian co-ordinates. For turbulent flow, Reynolds decomposition is commonly applied to decompose flow velocity into mean velocity ($\bar{u}$) and fluctuation ($u^{\prime }$), so that the instantaneous velocity becomes
(3)$$u\left(x,t\right)=\bar{u\left(x\right)}+u^{\prime }\left(x,t\right)$$

The RANS equation can be expressed as
$$\frac{\partial }{\partial t}\left(\rho \bar{u_{i}}\right)+\frac{\partial }{\partial x_{j}}\left(\rho \bar{u_{i}}\bar{u_{j}}\right)=-\frac{\partial \bar{p}}{\partial x_{i}}+\frac{\partial }{\partial x_{j}}\left[\mu \left(\frac{\partial \bar{u_{i}}}{\partial x_{j}}+\frac{\partial \bar{u_{j}}}{\partial x_{i}}\right)\right]+\frac{\partial }{\partial x_{j}}\left(\tau_{ij}\right)$$
where $\tau_{ij}=-\rho \bar{{u^{\prime }}_{i}}\bar{{u^{\prime }}_{j}}$ is known as the Reynold stress tensor. Based on Boussinesq’s hypothesis, the model for Reynold stress can be written as
$$\tau_{ij}=\mu_{t}\left(\frac{\partial \bar{u_{i}}}{\partial x_{j}}+\frac{\partial \bar{u_{j}}}{\partial x_{i}}\right)-\frac{2}{3}\rho k\delta_{ij}$$
where $k=\frac{1}{2}\bar{{u^{\prime }}_{i}}\bar{{u^{\prime }}_{j}}$ is the turbulent kinetic energy, $\mu_{t}$ is the turbulent eddy viscosity, and $\delta_{ij}$ is the Dirac delta function. In the RANS equation, the objective of Boussinesq’s hypothesis is to relate Reynolds stress to the mean velocity gradient. For this hypothesis, the turbulent kinetic energy and turbulent eddy viscosity models should be applied. 

To model turbulent flow at the post stenotic region in arteries, both the k-*ω* and k-*ε* models were used. For the k-*ω* model, turbulent kinetic energy *k* and specific dissipation rate *ω* were obtained from the following equations: $$\begin{array}{l} \frac{\partial k}{\partial t}+\frac{\partial k\bar{u_{j}}}{\partial x_{j}}=-\frac{1}{\rho }\left(\rho \bar{u_{i}}\;\bar{u_{j}}\right)\frac{\partial \bar{u_{i}}}{\partial x_{j}}-\beta^{*}k\omega +\frac{\partial }{\partial x_{j}}\left[\frac{1}{\rho }\left(\mu +\sigma^{*}\mu_{t}\right)\frac{\partial k}{\partial x_{j}}\right]\\ \frac{\partial \omega }{\partial t}+\frac{\partial \omega \bar{u_{j}}}{\partial x_{j}}=-\alpha_{1}\frac{\omega }{\rho k}\left(\rho \bar{u_{i}}\;\bar{u_{j}}\right)\frac{\partial \bar{u_{i}}}{\partial x_{j}}-\beta \omega^{2}+\frac{\partial }{\partial x_{j}}\left[\frac{1}{\rho }\left(\mu +\sigma \mu_{t}\right)\frac{\partial \omega }{\partial x_{j}}\right]\\ \mu_{t}=\frac{\rho k}{\omega } \end{array}$$

For the k-*ε* model, k and the dissipation rate *ε* were obtained from the following transport equations:$$\begin{array}{l} \frac{\partial \left(\rho k\right)}{\partial t}+\frac{\partial }{\partial x_{j}}\left(\rho k\bar{u_{i}}\right)=\frac{\partial }{\partial x_{j}}\left[\frac{\mu_{t}}{\sigma_{k}}\frac{\partial k}{\partial x_{j}}\right]+2\mu_{t}S_{ij}S_{ij}-\rho \varepsilon \\ \frac{\partial \left(\rho \varepsilon \right)}{\partial t}+\frac{\partial }{\partial x_{j}}\left(\rho \varepsilon \bar{u_{i}}\right)=\frac{\partial }{\partial x_{j}}\left[\frac{\mu_{t}}{\sigma_{k}}\frac{\partial \varepsilon }{\partial x_{j}}\right]+C_{1\varepsilon }\frac{\varepsilon }{k}2\mu_{t}S_{ij}S_{ij}-2C_{2\varepsilon }\rho \frac{\varepsilon^{2}}{k}\\ \mu_{t}=\rho c_{\mu }\frac{k^{2}}{\varepsilon } \end{array}$$

With this model, we could also define the generation of turbulent kinetic energy, $G_{k}=\mu_{t}S^{2}$, and $S=\sqrt{2S_{ij}S_{ij}}$ as the mean rate of strain tensor.

### 2.4. Physical Model Validation 

To validate the model and confirm the parameter settings, we compared our CFD simulation results using the patient-specific model after IVC. This model was used to simulate clinical findings that a poststenotic to prestenotic peak vein velocity ratio (defined as PVR in this study) of 2.5 is a critical value for diagnosing IVCS [4,5,39]. To test the effect of the PVR on flow status, the inferior vena cava outlet blood velocity (VI) was set to values ranging from 1.0 to 3.5 times the VL. Vortex patterns in the vessel past the compression site were observed and compared.

### 2.5. Problem Definition

In order to assess whether the tilt angles between the left and right common iliac veins play a role in IVCS, CT images of iliac veins with various angles were collected from 31 patients. Related patient variables are listed in Table 1. All CT images collected showed visible compression at the left iliac vein. These CT images (Figure 3a,d,g) were converted to solid models (Figure 3b,e,h). Finally, the patient-specific finite element models were meshed according to the method described above (Section 2.1) (Figure 3c,f,i).

The boundary conditions for all models were set according to previous models, i.e., with a PVR value of 2.5. The flow fields past the compression sites in the patient-specific iliac vein models were divided into three categories and scored. In this study, turbulent, vortexed, blocked, or obstructed blood flow patterns were defined as disturbed TVBO flow. Patient-specific models were scored 0, 1, or 2 points for laminar, transitional to turbulent, or TVBO, respectively. According to the classification of IVCS patients by Li et al. [3], the angle between the right or left common iliac vein and inferior vena cava at the confluence level of the common iliac vein were defined as α1 and α2 (Figure 1a). The proportion of turbulent or blocked flow between different angles was calculated and compared. In addition, the ratio of all disturbed flow, including transitional (1 point) and TVBO flow (2 points), was calculated according to the angle. Student *t*-tests were performed to evaluate the changes between various point groups. A *p*-value lower than 0.05 was considered statistically significant.

## 3. Results and Discussion

### 3.1. Material Properties Validation

The hemodynamic properties of blood, including the Newtonian fluid state, are key factors when building a mathematical model for the CFD analysis of the circulation system. Blood, which in larger vessels contains both liquid and solid parts, exhibits a Newtonian flow state and a constant viscosity that changes according to the vessel diameter. Blood plasma, which is an incompressible, Newtonian-homogeneous fluid that makes up over 50% of blood [33], exhibits Newtonian behavior [25]. In microcirculatory systems, however, such as small vessels and capillaries with diameters under 1 mm, blood can be treated as a non-Newtonian fluid [38] whose changing viscosity can be determined using a modified Quemada model [30]. Previous reports analyzing stenosis conditions have stated that assuming a non-Newtonian state may lead to better simulation results compared to a Newtonian model [25], because a non-Newtonian model is more appropriate for analyzing wall shear stress [40] and arterial stenosis in regions with Reynolds numbers below 100 [27,41]. However, since wall shear stress was not observed in this study, and because the Reynolds number of this research model was much greater than 100, we assumed our model to be Newtonian for all simulation procedures.

### 3.2. CFD Model Selection

Various mathematical models of turbulence have been developed, such as the k-ω and k-ε turbulence models. To test their calculation accuracy, these two turbulence models were used to measure flow disturbances past the compression site of the left common iliac veins. When a k-ω model was used to simulate the compressed iliac vein with IVCS in our simulation, the CFD model showed no significant disturbances in blood flow beyond the compression site (Figure 4a). However, significantly turbulent flow and obstruction were observed when the k-ε model was used (Figure 4b). This result suggests that the k-ε model used in the present investigation is a suitable Reynolds-averaged Navier–Stokes turbulence model that captures the turbulent transition and vessel obstruction found in compressed iliac veins. However, this finding is inconsistent with research into CFD analysis in stenosed vessels by Kabir et al. [38] which concluded that simulations using k-ω models are more realistic than those using the k-ε models, perhaps because their simulation was performed in blood vessels that exhibited non-Newtonian properties, rather than in the Newtonian fluid setting of the present study. In addition, studies have shown that the k-ω model is overly sensitive to turbulent fluids, and thus more suited to analyzing shear stress in near-wall regions [42], which falls beyond the scope of the present study.

A CFD model comprises three major components: governing equations, flow state, and boundary conditions, with flow state being most closely related to the material properties of the fluid. The Reynolds number (Re), the ratio of inertial to viscous forces within a fluid, is defined using fluid density, viscosity, and tube diameter and plays an important role in describing the flow state. Laminar flow is indicated by Re values below 2100, turbulent flow by values over 4000, and values between 2100 and 4000 indicate a transition state [43]. Since the diameter of blood vessels in the human body varies from small capillaries to large arteries, the Reynolds number varies from 1 in small arterioles to 4000 in the largest artery [44], model assumptions can affect mathematical simulation of blood flow different. Laminar flow has been assumed to give a reasonable approximation of blood flow through larger vessels [43] and has been applied in most analyses of stenotic flow and wall shear stress [32,38]. Re values calculated at the compression site in the present study fell between 2100 and 4000, meaning that our CFD model, rather than assuming laminar flow as in previous studies, needed to assume transitional flow. Since a steady laminar flow can become unsteady due to the presence of stenosis, transition to turbulence can occur even at a low Reynolds number [45]. As shown in Figure 4e, steady laminar flow was observed before the compression site. However, flow disturbances in the downstream region of the iliac vein model occurred after compression. Figure 4c shows an observable low-velocity pattern at the vortex area, which can be attributed to stenosis at the vessel compression site (Figure 4d).

### 3.3. Validating the CFD Model

Although stenosis in the iliac vein greater than 50% of the vein’s diameter is a clinical sign of IVCS [5,23], changes in the vein’s inner diameter can be difficult to measure in clinical settings. Additionally, even if CT venography clearly demonstrates iliac vein compression, imaging findings may only reflect a patient’s volume status rather than accurately confirming IVCS [9]. These difficulties make diagnosis based on diameter reduction a poor strategy [5]. In a study to determine criteria for clinically significant vein stenosis, Labropoulos et al. measured hemodynamic parameters in thirty-seven patients with signs and symptoms of central venous outflow obstruction using ultrasound. They reported a poststenotic to prestenotic peak vein velocity ratio (PVR) of 2.5 as the most important determiner of the presence of IVCS [5], and this clinical finding was used as a reference to validate the selected parameters used in the iliac vein model in the present study. When our model was simulated with different PVRs across stenosis lower than 2.0, no apparent disturbance in blood flow was observed (Figure 5a–c), while a peak vein velocity ratio of 2.5 across stenosis appeared to be the best cutoff for the presence of IVCS. When PVR values were higher than 2.5 (Figure 5d–f), a significant blood flow disturbance was observed, confirming previous findings [4,5,39]. To verify the governing equations, model geometry, and boundary conditions [43], a comparison of the results obtained by our mathematical model and clinical observations could be used for validation.

### 3.4. Patient-Specific CFD Simulation

To assess the compression characteristics of IVCS patients, Li et al. [3] established a new subtyping method for individuals with asymptomatic IVCS. According to findings in their study indicating that the dual compression of the left common iliac vein by the right common iliac artery and left internal iliac artery is the most common form of compression, the present study developed CFD models from CT images showing this pattern of compression.

The clinical prevalence of IVCS remains unknown and is difficult to determine [7,9,10]. The presence of iliac vein compression alone is insufficient for diagnosis [9], while approximately 24% of people in asymptomatic populations have significant compression at the left common iliac vein [7]. The incidence of symptomatic patients with deep vein thrombosis of the left leg was reported to vary from 18 to 49% [15]. When the boundary condition in our study was set as PVR 2.5 before and after the compression site, the disturbance flow field after compression at the left common iliac vein was 38.7%, a value that confirmed clinical reports. Interestingly, blood flow obstruction in our simulation occurred at a rate of 9.7% (3/31), a result similar to a previous clinical survey in which only 9% of left IVC patients had a degree of compression greater than 50% [3]. 

Another study measuring major iliac vein angles in fifty patients without venous disease by Song et al. found that the left side pathway was more complex than the right, leading the authors to conclude that the iliac vein angle is important in the planning of iliac vein stenting therapy [20]. In the present study, the angles between the common iliac veins and inferior vena cava at the confluence area were measured. The average angle between the left and right iliac veins (α1 + α2) of 83.0 ± 22.6 degrees was lower than the average measurements by Song et al. (α angle = 147.6 degrees), because of the difference in the angle of view used. The angles between the left and right iliac veins were divided into α1 and α2 in this study according to the classification of Li et al. [3]. The α2 angle (48.8 ± 17.7 degrees) was significantly larger (*p* < 0.01) than the α1 angle (34.1 ± 11.9 degrees). This result was consistent with Li et al.’s clinical survey, which also found that the left iliac vein angle was significantly larger than the right side angle [3]. This physiological characteristic may contribute to the higher prevalence of left common iliac vein compression [3]. Although researchers have reported that blood flow is laminar, flow can shift from transitional to turbulent flow or create a vortex due to the changing blood hemodynamics leading up to vascular diseases [46]. All four types of blood flow were observed in our simulation using patient-specific models (Figure 6).

Figure 7 shows the tendency towards increasing flow field disturbance with increasing α2 angles (R = 0.995) (Figure 7b). This tendency was not, however, found for α1 angles. When the α1 angle was less than 25 degrees, 25–40 degrees, or greater than 40 degrees, the incidence of blood flow disturbance past the compression site was 50%, 70%, and 66.7%, respectively (Figure 7a). When the α2 angle was lower than 40 degrees, blood flow disturbance occurred at a rate of 33.3%, which increased to 58.3% when the α2 angle fell between 40 and 60 degrees (Figure 7b). The incidence of blood flow disturbance reached 90% at α2 angles larger than 60 degrees. For α1 + α2, the incidence of blood flow disturbance rates were 44.4%, 54.5%, and 81.8% for angles lower than 70 degrees, 70–90 degrees, and greater than 90 degrees, respectively (Figure 7c). These results suggest that the α2 angle, but not α1, can be used as an indicator of IVCS.

Since IVCS is caused by both mechanical and physiologic factors [9], it is not surprising that the degree of flow disturbance depends on both stenosis severity and vessel geometry [38]. As discussed above, the location of iliac vein compression affects IVCS severity. Since the left and right common iliac veins are asymmetrical, different tilt angles are attributed to different levels and depths of compression, which leads in turn to different symptoms [3]. Although several methods, such as ultrasound imaging, are used for diagnosing lower limb disorders, even experienced professionals have a 20% chance of misdiagnosing IVCS [39]. In this study, no significant difference in flow type was found in compressed iliac veins according to the α1 angle (Figure 8a). The average angle value for vessels with laminar, transitional, and TVBO flow were 39.0 ± 13.0 degrees, 51.6 ± 17.6 degrees, and 58.4 ± 17.6 degrees, respectively (Figure 8b). When the α2 angle was used as a classification tool, a significant difference in angle was found between compressed veins with laminar and transitional flow, as well as between laminar and TVBO flow (*p* < 0.05). Similar results were found for α1 + α2 (*p* < 0.01) (Figure 8c). These results confirmed the findings of Li et al., who suggested that vein compression could contribute to the tilt angle between the common iliac vein and inferior vena cava [3] and suggested that the α2 or α1 + α2 angle could both predict IVCS and reflect the severity of disturbance in the vessel. They concluded that the α2 angle is a useful reference to predict IVCS and the condition’s severity and to assist clinical decision making.

### 3.5. Study Limitations

As emphasized by Kamada et al., blood flow models using CFD cannot perfectly reproduce blood flow in vivo [26]. The main limitation of this study was that several boundary conditions needed to be set in order to simplify the mathematical model. For example, this study assumed that the vessel walls in the patient-specific model were rigid, an assumption that led to minor differences in simulation results due to wall shear stress and the relative velocity of the wall, because blood in contact with the wall was not discussed, as in a previous study [30]. A rigid wall assumption has been included in the majority of computational flow studies [47], such as a 2022 investigation of the CFD simulation of the cardiovascular system by Albadawi et al. [33], which concluded that assuming a rigid wall led to only slight differences when modeling large-diameter vessels. Since the iliac vein is relatively large in diameter, assuming a rigid wall seemed acceptable for the present model. Another possible limitation of the present study was the effect of the heartbeat on the great vessels; however, since the object of this study was a vein, vascular pulsation caused by the beating heart had little effect on the simulation results. Thus, in spite of these disadvantages, the CFD model developed in this study provides useful information for investigating IVCS.

## 4. Conclusions

The present study indicated that as the angle between the left and right common iliac veins increases, so does the prevalence of IVCS. Our findings also suggested that the peak velocity ratio across the compressed site (PVR > 2.5) and the angle between the left and right iliac vein can be used in combination as a diagnostic reference to determine the presence of IVCS. Since no standardized clinical diagnostic guideline to identify IVCS currently exists [9], these results can be used to reduce the misdiagnosis and inappropriate treatment of IVCS to a considerable extent.

## Figures and Tables

**Figure 1 bioengineering-10-00688-f001:**
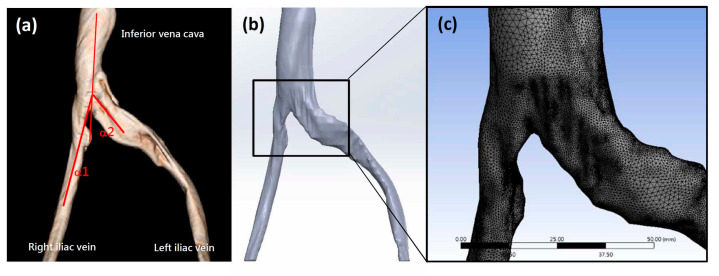
(**a**) A typical three-dimensional reconstructed iliac vein CT image. (**b**) The image was converted to a solid digital model. (**c**) After the meshing process, a finite element model was created using commercial software.

**Figure 2 bioengineering-10-00688-f002:**
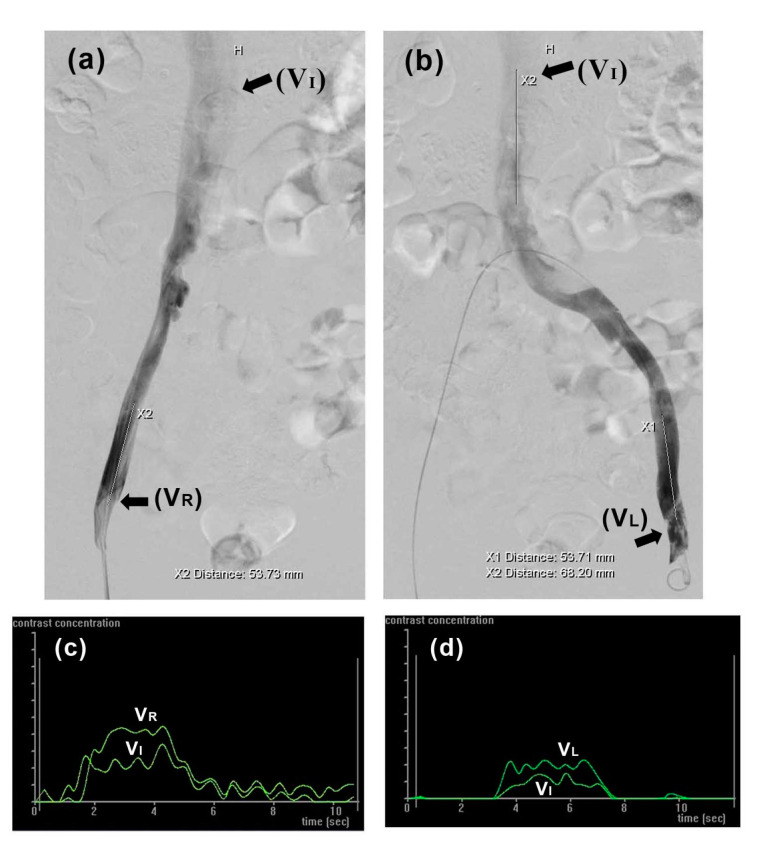
Blood flow velocity in iliac veins was determined by intravenous digital subtraction angiography (IVDSA). (**a**,**b**) Contrast agent propagating through right and left iliac veins, respectively; (**c**,**d**) corresponding time intensity curves used for calculating VI, VR, and VL.

**Figure 3 bioengineering-10-00688-f003:**
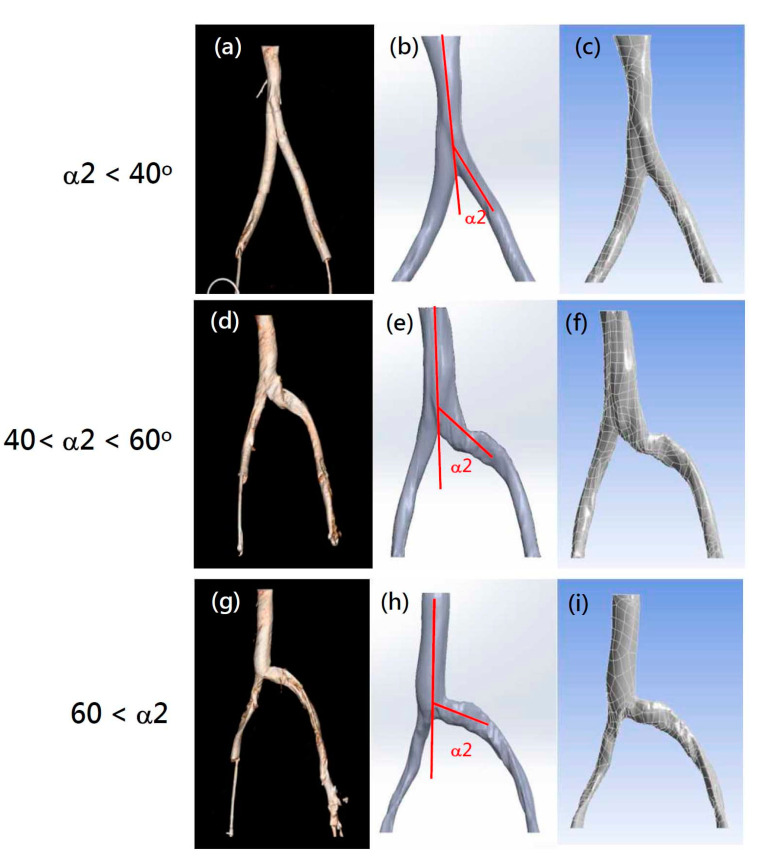
Examples of selected patient CT images (**a**,**d**,**g**), digital solid models (**b**,**e**,**h**), and finite element CFD models (**c**,**f**,**i**) with various α2 angles. (**a**–**c**) Models with α2 angle under 40 degrees; (**d**–**f**) analog images with α2 angles between 40 and 60 degrees; (**g**–**i**) veins with α2 angles above 60 degrees.

**Figure 4 bioengineering-10-00688-f004:**
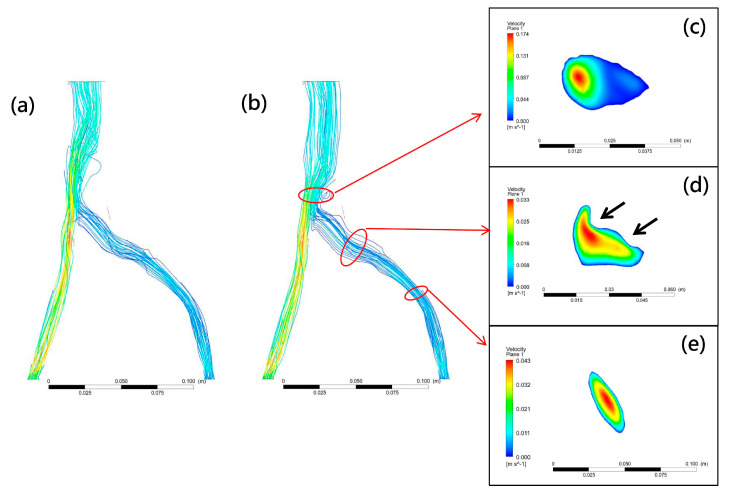
Typical CFD model flow fields calculated using kω (**a**) and kε (**b**) turbulence models. Vortex occurs past compression site with an extremely low blood velocity ((**c**), blue contour). For compression (**d**) and non−compression sites (**e**), high−speed flow (red contour) surrounded by low-speed flow (blue contour) was observed in the central vessel area. This steady laminar flow can be observed at compression and non-compression sites. Black arrows indicate compression site.

**Figure 5 bioengineering-10-00688-f005:**
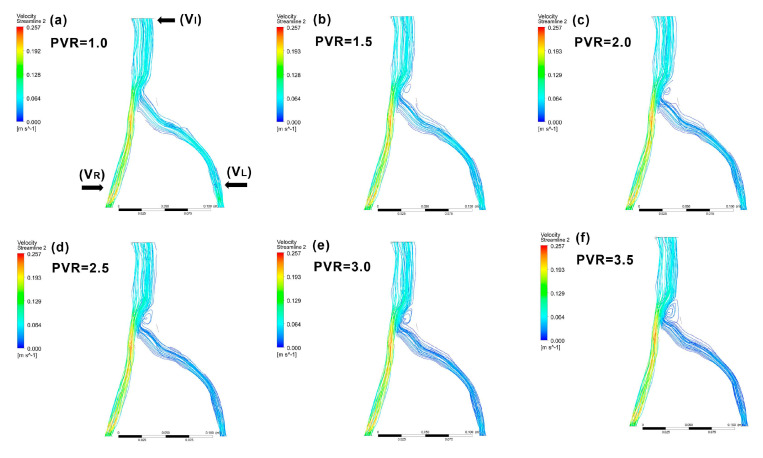
Examples of patient-specific CFD models with various poststenotic to prestenotic peak vein velocity ratios (PVRs). A PVR of 2.5 across the stenosis is the best diagnostic reference for IVCS. When the model was simulated with PVRs across the stenosis under 2.0, no apparent blood flow disturbance was observed (**a**–**c**). At PVRs above 2.5 (**d**–**f**), significant blood flow disturbance occurred. PVR: peak vein velocity ratio.

**Figure 6 bioengineering-10-00688-f006:**
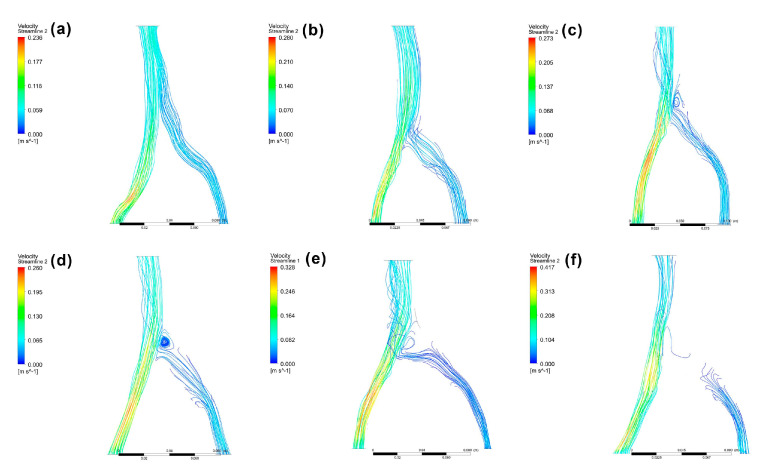
CFD model examples with laminar flow (**a**), transitional flow (**b**), turbulent flow (**c**), vortex (**d**), block (**e**), and obstruction (**f**). Scores of 0, 1, and 2 were attributed to models (**a**,**b**,**c**−**f**), respectively.

**Figure 7 bioengineering-10-00688-f007:**
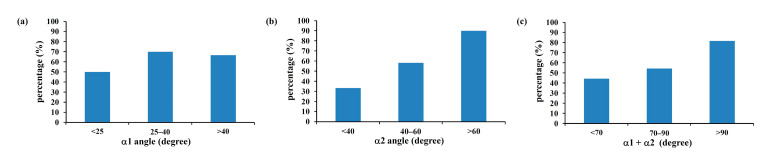
The incidence of blood flow disturbance past the compression site of the CFD models according to α1 (**a**), α2 (**b**), and α1 + α2 (**c**). Although the incidence of blood flow disturbance showed no significant change when using α1 angles (**a**), an effect was obvious when α2 (**b**) and α1 + α2 (**c**) were considered.

**Figure 8 bioengineering-10-00688-f008:**
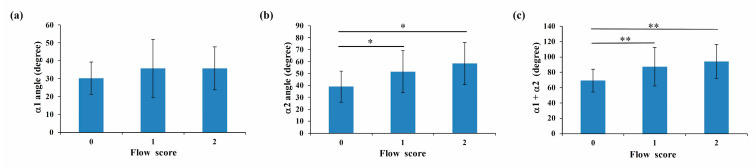
The average angle values of α1 (**a**), α2 (**b**), and α1 + α2 (**c**) for vessels with laminar (score 0), transitional (score 1), and TVBO flow (score 2). Significance is indicated as * (*p* < 0.05) and ** (*p* < 0.01).

**Table 1 bioengineering-10-00688-t001:** Patient variables.

Variable	
Age, years	58.19 ± 15.45 (21–79)
Male	11 (35.5%)
Female	20 (64.5%)
α1, degrees	34.1 ± 11.9 (18–65)
α2, degrees	48.8 ± 17.7 (25–90)
α1 + α2, degrees	82.9 ± 22.8 (45–130)

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
