# Peer review of "CFD Study of the Effect of the Angle Pattern on Iliac Vein Compression Syndrome"

_bioengineering, 2023, doi:10.3390/bioengineering10060688_

Round 1
Reviewer 1 Report
In the introduction, authors present a satisfactory explanation of all the concepts needed to the presented work as well as the description of other studies developed in the area. All the limitations of the numerical simulations performed in the work were presented and discussed in a very clear way.
Numerical simulations complemented with clinical data may be the answer to diagnose asymptomatic diseases and the authors do it in this paper.
The numerical simulations developed by the authors were very well described and figures represent clearly the geometry and used mesh. Additionally, authors refer the essential steps for the validation of the used numerical model reproducing real iliac veins however no quantitative comparison between numerical results and clinical observations were presented. Could the authors do this comparison?
Author Response
Bioengineering:
Dear Editor,
Thank you very much for handling our manuscript “The effect of angle pattern on iliac vein compression syndrome: An analysis using computational fluid dynamics” (bioengineering-2419400). We would like to thank the reviewers for cautiously reviewing the manuscript and providing valuable comments and suggestions, which is helpful to improve the quality of our study. We have clarified these issues in the revision, with the additions and amendments in the manuscript. A detailed point-by-point response is given below. To answer the comments from the reviewers, red words with yellow background were used in the revised manuscript. For re-phrase the sentence, Track Changes” function was used.
Sincerely,
Haw-Ming Huang
School of Dentistry,
Taipei Medical University, Taipei, Taiwan
Reviewer 1:
Comment 1.1: In the introduction, authors present a satisfactory explanation of all the concepts needed to the presented work as well as the description of other studies developed in the area. All the limitations of the numerical simulations performed in the work were presented and discussed in a very clear way. Numerical simulations complemented with clinical data may be the answer to diagnose asymptomatic diseases and the authors do it in this paper.
Author Response: We sincerely thank the reviewer for his comments.
Comment 1.2: The numerical simulations developed by the authors were very well described and figures represent clearly the geometry and used mesh. Additionally, authors refer the essential steps for the validation of the used numerical model reproducing real iliac veins however no quantitative comparison between numerical results and clinical observations were presented. Could the authors do this comparison?
Author Response: We thank the reviewer for this comment. In the revised manuscript, we added new paragraphs to include quantitative comparison between numerical results and clinical observations on page 10, second paragraph as follows: “Interestingly, blood flow obstruction in our simulation occurred at a rate of 9.7% (3/31), a result similar to a previous clinical survey in which only 9% of left IVC patients had a degree of compression greater than 50% [3].

Reviewer 2 Report
The authors present an interesting and useful study of the effect of angle pattern on the iliac vein compression syndrome. They should correct language errors, improve the narrative and clarity of the text and address the indicated technical deficiencies. Some improving suggestions are as follows:
01. Lines 2-3: The authors may wish to consider the more concise title: “CFD study of the effect of angle pattern on the iliac vein compression syndrome”.
02. Line 43: Better to use as keyword “angle pattern” instead of simply “angle”.
03. L78-79: The statement “Velocity, volume, and pressure of blood flow before and after the vessel have shown promise” seems to be inconclusive.
04. L89: Better to write “Liu et al. [23]”. L93: Better to write “Li et al. [3]”.
05. L101: Write “Hemodynamics can be studied …” instead of “Hemodynamics can be calculated …”
06. L117: Write simply “Li et al. [3] …” instead of “Li et al. (2021) … [3]”.
07. L133: Write “iliac vein” instead of “lilac vein”.
08. L143: It would be better to use a different section “3. Problem definition and modelling”. Then add a new subsection “3.1 Problem definition”.
09. L158-159: Delete this sentence as a redundant repetition.
10. L165: Before this line, add a subsection heading “3.2 Mathematical modelling”.
11. L165-200: Rephrase this text to improve its narrative and clarity.
12. L165-168: Better to rephrase this sentence as: “Blood flow passing through a stenosis may produce turbulence with characteristics depending on the degree of stenosis. Thus, it is important to use an appropriate turbulence model to calculate the flow field and obtain the hemodynamic parameters”.
13. L172-173: Rephrase this sentence as: “The Reynolds-averaged Navier-Stokes (RANS) equations were used as flow governing equations. Firstly, the continuity and Navier-Stokes (NS) equations are:”.
14. L179: Write “decompose” instead of “separate”.
15. L189: Rephrase to “…hypothesis is to relate the Reynolds stress…”
16. L193: Write “models were used” instead of “models are studied”.
17. L207-213: Rephrase this paragraph to improve its narrative and clarity. To distinguish between the physical and CFD models, use as heading “Physical model validation” instead of “Model validation”.
18. L212-213: Add a figure to support the statement “The vortex patterns in the vessel past the compression site were observed and compared”.
19. L220: Place Fig. 3 close to where it is cited, then present it in detail.
20. L235-249: Rephrase this paragraph to improve its narrative and clarity.
21. L251-266: Rephrase this paragraph to improve its narrative and clarity.
22. L266-267: Improve the clarity and description of Fig. 4.
23. L271-291: Rephrase this paragraph to improve its narrative and clarity.
24. L294-312: Rephrase this paragraph to improve its narrative and clarity.
25. L312-313: Improve the clarity of Fig. 5.
26. L330: Write “Song et al. [20]”.
27. L330-346: Rephrase this paragraph to improve its narrative and clarity.
28. L346-347: Improve the clarity of Fig. 6.
29. L351-362: Rephrase this paragraph to improve its narrative and clarity.
30. L346-347: Improve the clarity and caption of Fig. 7.
31. L366-367: Write “it is not surprising” instead of “it is unsurprising”.
32. L397: Write “Albadawi et al. [33]”.
33. L389-404: Add the non-Newtonian nature of blood as an additional limitation.
The authors should correct the language errors, and improve the narrative and clarity of the text.
Author Response
Bioengineering:
Dear Editor,
Thank you very much for handling our manuscript “The effect of angle pattern on iliac vein compression syndrome: An analysis using computational fluid dynamics” (bioengineering-2419400). We would like to thank the reviewers for cautiously reviewing the manuscript and providing valuable comments and suggestions, which is helpful to improve the quality of our study. We have clarified these issues in the revision, with the additions and amendments in the manuscript. A detailed point-by-point response is given below. To answer the comments from the reviewers, red words with yellow background were used in the revised manuscript. For re-phrase the sentence, Track Changes” function was used.
Sincerely,
Haw-Ming Huang
School of Dentistry,
Taipei Medical University, Taipei, Taiwan
Reviewer 2
Comment 2.1: Lines 2-3: The authors may wish to consider the more concise title: “CFD study of the effect of angle pattern on the iliac vein compression syndrome”.
Author Response: We thank this commend from the reviewer. The title of the revised manuscript was changed to “CFD study of the effect of angle pattern on the iliac vein compression syndrome”.
Comment 2.2: Line 43: Better to use as keyword “angle pattern” instead of simply “angle”.
Author Response: We thank this the reviewer for this comment. The keyword “angle” was revised to “angle pattern”.
Comment 2.3: L78-79: The statement “Velocity, volume, and pressure of blood flow before and after the vessel have shown promise” seems to be inconclusive.
Author Response: We thank the reviewer for this comment. This inconclusive statement was removed in the revised manuscript.
Comment 2.4: L89: Better to write “Liu et al. [23]”. L93: Better to write “Li et al. [3]”.
Author Response: The citation of Liu et al. and Li et al. were revised according to the reviewer’s comment.
Comment 2.5: L101: Write “Hemodynamics can be studied …” instead of “Hemodynamics can be calculated …”
Author Response: The statement “Hemodynamics can be calculated …” was revised to “Hemodynamics can be studied…”.
Comment 2.6: L117: Write simply “Li et al. [3] …” instead of “Li et al. (2021) … [3]”.
Author Response: L117: The citation was revised to Li et al. [3].
Comment 2.7: L133: Write “iliac vein” instead of “lilac vein”.
Author Response: We thank the reviewer for pointing out this typo error. On L133: “lilac vein” was revised to “iliac vein”.
Comment 2.8: L143: It would be better to use a different section “3. Problem definition and modelling”. Then add a new subsection “3.1 Problem definition”.
Author Response: Due to the format regulation of the Journal, we did not add a new section 3. However, we revised the subsection 2.4 as “2.4 Problem definition”.
Comment 2.9: L158-159: Delete this sentence as a redundant repetition.
Author Response: Thank you for this comment. The sentence in L158-159 is deleted in the revised manuscript.
Comment 2.10: L165: Before this line, add a subsection heading “3.2 Mathematical modelling”
Author Response: According to the reviewer’s comment, we added a new subsection heading “2.3 Mathematical modelling” on page 5.
Comment 2.11: L165-200: Rephrase this text to improve its narrative and clarity.
Author Response: According to the reviewer’s comment, we added a new subsection heading “2.3 Mathematical modelling” on page 5.
Comment 2.12: L165-168: Better to rephrase this sentence as: “Blood flow passing through a stenosis may produce turbulence with characteristics depending on the degree of stenosis. Thus, it is important to use an appropriate turbulence model to calculate the flow field and obtain the hemodynamic parameters”.
Author Response: The sentences from L185-168 have been revised according to the reviewer’s comment.
Comment 2.13: L172-173: Rephrase this sentence as: “The Reynolds-averaged Navier-Stokes (RANS) equations were used as flow governing equations. Firstly, the continuity and Navier-Stokes (NS) equations are:”.
Author Response: The sentences from L172-173: have been revised according to the reviewer’s comment.
Comment 2.14: L179: Write “decompose” instead of “separate”.
Author Response: L179: “separate” was revised to “decompose”.
Comment 2.15: L189: Rephrase to “…hypothesis is to relate the Reynolds stress…”
Author Response: L189: The revised manuscript has been rephrased to “…hypothesis is to relate the Reynolds stress….
Comment 2.16: L193: Write “models were used” instead of “models are studied”.
Author Response: L193: “models are studied” was revised to“models were used”.
Comment 2.17: L207-213: Rephrase this paragraph to improve its narrative and clarity. To distinguish between the physical and CFD models, use as heading “Physical model validation” instead of “Model validation”.
Author Response: We thank the reviewer for this comment. The paragraph in the manuscript has been revised as, “To validate the model and confirm setting parameters… patterns in the vessel past the compression site were observed and compared.” In addition, the heading “Model validation” was revised to “Physical model validation”.
Comment 2.18: L212-213: Add a figure to support the statement “The vortex patterns in the vessel past the compression site were observed and compared”.
Author Response: We thank the reviewer for this comment. The figure was shown in Fig. 5.
Comment 2.19: L220: Place Fig. 3 close to where it is cited, then present it in detail.
Author Response: In the revised manuscript. Fig. 3 was moved closer to the cited text. The detailed description of Fig. 3 was added on page 7, first paragraph.
Comment 2.20, 21, 23, 24, 27, 29: L235-362: Rephrase these paragraphs to improve its narrative and clarity.
Author Response: We thank the reviewer for these comments. These paragraphs have been rephrased to improve narrative and clarity in the revised manuscript using “Track Changes” function.
Comment 2.22, 25, 28, 30: Improve the clarity of Fig.4-Fig. 7
Author Response: The resolution of figure 4-7 fit the regulation and rule of the Journal. However, we found the resolution was reduced when the figure was transferred to PDF format during the reviewing stage. The image will return to normal resolution after formatting for publishing.
Comment 2.22. L266-267: Improve the description of Fig. 4.
Author Response: In the revised manuscript, more description was added to the caption of Fig. 4.
Comment 2.26: L330: Write “Song et al. [20]”
Author Response: L330: The citation was revised to “Song et al. [20]”
Comment 2.30. L346-347: Improve the caption of Fig. 7.
Author Response: In the revised manuscript, more description was added to the caption of Fig. 7.
Comment 2.31: L366-367: Write “it is not surprising” instead of “it is unsurprising”.
Author Response: The phrase “it is unsurprising” was revised to“it is not surprising”
Comment 2.32: L397: Write “Albadawi et al. [33]”.
Author Response: The citation was revised to “Albadawi et al. [33]”
Comment 2.33: L389-404: Add the non-Newtonian nature of blood as an additional limitation.
Author Response: The limitation of non-Newtonian nature of blood was explained on page 8, section 3.1.
